# Risk Factors for Upper Limb Injury in Tennis Players: A Systematic Review

**DOI:** 10.3390/ijerph17082744

**Published:** 2020-04-16

**Authors:** Afxentios Kekelekis, Pantelis Theodoros Nikolaidis, Isabel Sarah Moore, Thomas Rosemann, Beat Knechtle

**Affiliations:** 1Department of Sports Science, Cardiff Metropolitan University, Cardiff CF23 6XD, UK; akekelekis@gmail.com (A.K.); imoore@cardiffmet.ac.uk (I.S.M.); 2Sport Injury Clinic for Prevention and Rehabilitation, 72100 Agios Nikolaos, Greece; 3Exercise Physiology Laboratory, 18450 Nikaia, Greece; pademil@hotmail.com; 4School of Health and Caring Sciences, University of West Attica, 12241 Athens, Greece; 5Institute of Primary Care, University of Zurich, 8091 Zurich, Switzerland; thomas.rosemann@usz.ch; 6Medbase St. Gallen Am Vadianplatz, 9001 St. Gallen, Switzerland

**Keywords:** kinetics, kinematics, scapular dyskinesis, muscle fatigue, prolonged tennis (exposure)

## Abstract

Studies in tennis injuries have successfully identified the incident rate, the location, and the type of the injury. The majority of the studies have multiple perspectives (epidemiology, biomechanics, performance), however only a few studies were able to identify risk factors or mechanisms that contribute to tennis injuries. Until now, there has not been a systematic literature review that identifies risk factors for tennis injuries. The objective of this review was to identify and critically appraise the evidence related to risk factors for upper limb injury in tennis players. A systematic review was conducted in accordance with the Preferred Reporting Items for Systematic Reviews and Meta-Analyses (PRISMA) framework, using a research question developed by the Patient Problem, (or Population) Intervention, Comparison or Control, and Outcome (PICO) methodology. The quality of the studies included was moderate to low, indicating prolonged tennis (exposure to tennis), scapular dyskinesis, muscle fatigue, scapulothoracic properties, shoulder kinetics or kinematics, skill level, and technique as risk factors for upper limb injury in tennis players. In this review, it is evidenced that the majority of tennis injuries are associated with overuse and a chronic time course, however, tennis injuries do not arise from a linear combination of isolated and predictive factors. Therefore, the multifactorial and complex nature of tennis injuries has to be further examined. The necessity of more randomized control trial studies is highly recommended.

## 1. Introduction

Participation in tennis places players at risk of injury—independently of performance level (i.e., competitive or recreational)—while tennis injury profile remains unique in comparison to other racquet and overhead sports, regarding physical demands, biomechanical loads, and equipment [1]. Tennis differs from other sports in terms of match duration (exposure), surface of play, and equipment [2] and is characterized by high-velocity repetitive upper limb movements, leading to overuse injuries, while sprinting, stopping, jumping, landing, and pivoting place high linear and rotational loading forces onto the joints of the lower extremity, increasing the risk for acute injury [3]. Data from one of the most recent epidemiological studies on tennis injuries reported upper limb injuries to account for 28% of all injuries for male adult players and 23% for female players, while the shoulder joint was reported to be the most frequently injured site of the upper limb [4]. 

Injuries to the shoulder joint are very frequent in professional tennis players, mainly due to the repetitive mechanical overload of the shoulder joint [5]. Shoulder pain was present in 24% of elite tennis players from age 12–19 years, with the prevalence increasing to 50% for senior ex-professional tennis players (over 35 years old) [6]. In throwing sports, alterations in scapular position and motor control have been reported to account 67%–100% of shoulder injuries, including rotator cuff tears, impingement, and glenohumeral instability [7]. Alterations in periscapular muscle properties could cause posterior shoulder tightness and further glenohumeral internal rotation deficit (GIRD), which has been defined as one of the most frequent rotational adaptations of the shoulder joint. GIRD has been defined as the loss in degrees of glenohumeral internal rotation of the throwing shoulder compared to the non-throwing shoulder [8]. It is believed that GIRD is one of the risk factors behind a shoulder injury amongst tennis players [8]. Tendinopathy was documented to be the most frequent type of injury to the wrist since tennis players add a lot of spin and speed on their strokes [2] and wrist tendinopathy has been reported to have the highest incident rate of tendon injuries in a 15-year study on the epidemiology of injuries at the US OPEN Grand Slam tournament [9].

The term “kinetic chain” is used in order to describe the synchronous use of selective muscle groups, segmental rotations, and coordinated lower extremity muscle activation that transfer the lower body force production through the core to the upper body and out through the racket into the ball [8,10]. Kovacs and Ellenbecker [10] describe the tennis serve as the most complex stroke in competitive tennis and it is suggested that the serve is one of the most injurious of all tennis strokes, due to the repetitive nature and the overhead motion, which adds tremendous force on the torso and upper extremity [11].

Studies in tennis injuries have identified the incident rate, the location, and the type of injury [4,9,12]. However, only a few studies were able to identify risk factors or mechanisms that contribute to tennis injuries [13,14]. Until now, there has not been a systematic literature review that identifies risk factors for upper limb injuries in tennis players. The objective of this review was to identify and critically appraise the evidence related to risk factors for upper limb injury in tennis players.

## 2. Experimental Section

This systematic review was conducted in accordance with the framework provided by Preferred Reporting Items for Systematic Reviews and Meta-Analyses (PRISMA) statement [15], using a research question developed with the Patient Problem, (or Population) Intervention, Comparison or Control, and Outcome (PICO) methodology (Figure 1). Articles were identified by searching the following databases MEDLINE, CINAHL, and SPORTDiskus from the 15th to the 20th of January 2017, using Medical Subject Headline (MeSH) terms in various combinations as listed on Table 1. Combinations between MeSH terms were applied, with the aim of identifying hidden studies. The articles were organized using reference management software package, EndNote V.8 (Thomson Reuters).

### 2.1. Study Selection

Research studies were included if they investigated the association between any potential risk factor and upper limb injury in tennis players. Articles were included if they met the following criteria: (1) they contained original data, (2) analytic design (cross-section study, prospective study, or randomized control trial, or laboratory study), (3) the study participants included were tennis players of any age, gender, and skill, (4) identify of at least one possible criterion that may be associated with upper limb injury, (5) be available in full text, (6) written in English language and (7) be published after 2000. Only studies that have been published in the last 17 years were included because the game of tennis has changed. It has become more powerful, more aggressive, and more tournaments have been added to the annual tennis calendar [16]. These characteristics may confound, resulting in different outcomes relevant to outcomes of studies conducted during the 1980s or even the 1990s. Full text articles were retrieved if their abstract provided insufficient data to confirm eligibility.

### 2.2. Data Extraction 

The following data were extracted from each study: authors, study design, population, sample size, and risk factors that were examined. Quality and level of evidence for individual studies were assessed. Quality of evidence was evaluated based on criteria for internal validity (study design, quality of reporting, presence of selection and misclassification bias, potential confounding) and external validity, using the Downs and Black (DB) quality assessment tool. The DB criteria assigns an individual score calculated of 27 total points for each study (10 points for reporting, 3 points for external validity, 7 points for bias, 6 points for confounding and 1 point for power [17]. The level of evidence represented by each study was categorized based on a modification of the Oxford Centre of Evidence-Based Medicine (OCEBM) 2009 model [18]. The 2009 model of the OCEBM was chosen over the 2011 model as it more sufficiently demonstrated the distribution of the tennis injury literature with regard to evidence hierarchy. It has also been used in a previous systematic review on risk factors for musculoskeletal injury in professional dancers [19]. As per study exclusion criteria, levels 1a, 2a, 3a (systematic reviews), 4 (case-series), and 5 (opinion-based papers) were excluded. The author resolved discrepancies in DB scores or OCEBM categorization.

## 3. Results

An overview of the study identification process is provided in Figure 1. The initial search yielded 17,431 articles (8044 from PubMed, 5028 from SPORTDiscus, 2673 from MEDLINE, 1679 from CINAHL and 7 from other sources). After duplicates were removed, 2622 potentially relevant articles remained for further procedures. Following the removal of studies not meeting inclusion criteria based on abstract review (articles were not relevant to tennis injuries and any potential risk factor for tennis injury), 144 articles remained for assessing eligibility. A total of 124 articles were excluded for the following reasons: the title and the content of the abstract were not relevant (*n* = 80); failed to identify risk factor for upper limb injury (*n* = 18); examining potential tennis injuries according to their anatomical location (*n* = 5). A further nine articles did not meet the study design inclusion criteria, and therefore were excluded because they were epidemiological studies, six articles were review studies, two articles were computer stimulation studies, two articles case studies, one case-control study, one editorial and one case-study. Finally, 20 articles were included in this systematic review.

Characteristics of the twenty articles included in this review are summarized in Table 2. The pooled sample was 961 participants (681 male and 280 female athletes) including 640 adults, 293 juniors, and 28 seniors. Of these participants, 11.2% of them were professional tennis players, 16.9% elite junior players, 3.7% college tennis players, and 68.2% were recreational tennis players. There were nine laboratory-based studies, eight cross-sectional studies, one cohort study, one randomized controlled clinical trial, and one prospective two-year follow-up study. The participation sample of the studies was tennis players from different ages, genders, and skill levels. The majority of the studies used a cohort of only one skill level, except one study, which examined potential risk factors between professional and advanced tennis players.

On the basis of the Downs and Black (DB) criteria, the median methodological quality of all 20 studies was 12 out of 33 (range 5–15). From all studies included, the randomized control clinical trial study and the cohort study received the highest scores, 15/33 and 14/33, respectively. One study failed to clearly describe the hypothesis of the research question, while three studies failed to clearly describe the research outcomes and the participant’s characteristics. A detailed summary of the checklists for measuring quality of study is listed in Table 3, Table 4, Table 5 and Table 6. The highest level of evidence demonstrated by all reviewed studies was level 2b (randomized controlled clinical trial and cohort studies). About 30% of the studies (*n* = 7) were classified as level 3 (cross-sectional study design and laboratory studies). The quantity, quality and level of evidence for the risk factors associated with upper limb injury in tennis players are synopsized in Table 7.

### Key Findings—Risk Factors 

The most common risk factor investigated was prolonged tennis, since five studies examined different effects of prolonged tennis to upper limb anatomical, structural, and biomechanical properties. Exposure to tennis is connected to tennis injuries, affecting shoulder muscle performance, causing decreased serve maximal angular velocities and joint kinetics [20]. Skill level and technique have been identified as risk factors for upper limb injury in tennis. Shoulder loading, especially during the serve and the smash is directly connected to the level of players’ skill. Professional tennis players are able to place lower loads to the shoulder joint kinetics decreasing the percentages developing shoulder, elbow, or wrist injury [21]. Shoulder flexibility and previous injury were investigated from two studies respectively. Cools and colleagues [22] and Kibler and Chandler [23] investigated the impact of tennis play on shoulder flexibility, especially the shoulder internal rotation. Both studies revealed that tennis players present significant changes in the dominant shoulder rotational properties. Several studies identified risk factors relative to the racquet properties [24], racquet grip [25], and racquet inertia [26], while a limited number of studies focused on shoulder and scapular kinematics and kinetics. Only one study investigated the relationship between age and tennis injury in the upper extremity [27]. The highest level of evidence (Level 2) was provided from two studies; one cohort study that examined the effect of prolonged tennis on the glenohumeral rotation and one randomized control trial, which examined the influence of fatigue on scapular kinematics. The remaining studies were lower quality studies (Level 3), eight cross-sectional studies, nine laboratory studies and, one prospective two-year follow-up study.

## 4. Discussion

This is the first systematic review to evaluate risk factors for upper limb injury in tennis players through critical appraisal of the literature. A total of 14 risk factors have been investigated, including prolonged tennis, racquet properties, shoulder and scapular kinematics, previous injury, shoulder flexibility, skills, technique, shoulder muscle properties, shoulder and scapular asymmetry, muscle fatigue, scapulothoracic position, glenohumeral instability, age, and kinetic chain integrity. Despite the quantity of risk factors investigated, the overall quality and level of evidence was low to moderate. The highest level of evidence (Level 2) was reported from only two studies, a cohort study, which examined the effect of prolonged tennis on glenohumeral rotation [28], and a randomized control trial study that examined the influence of fatigue on scapular kinematics [29]. Only three studies used professional tennis players.

### 4.1. Prolonged Tennis (Exposure to Tennis)

Tennis differs from other sports in terms of match duration (exposure) [2]. A tennis match can be played over several hours (1–5 h). Therefore, the term “prolonged tennis” is used to explain the nature of the sport in terms of exposure time and duration.

Prolonged tennis affects the glenohumeral rotational properties [28]. Jayanthi and colleagues [14] documented that volume of matches played increased the risk of injury within a tournament, especially after the fourth match played. Moore-Reed and colleagues [28] reported changes in key components of glenohumeral motion, including decreased glenohumeral rotation and shoulder total range of motion, decreased shoulder strength, and increased muscle stiffness 24 hours after playing tennis; variables that may contribute to shoulder and elbow injury. This study failed to provide evidence regarding the factors responsible for the variability in the changes, including exposure, muscle properties, kinetic chain alterations, or training volume. Further, the sample size was limited to only female participants and the study only examined the acute effects of prolonged tennis to GIR in field environment for only 24 hours post-game.

A high level of evidence (Level 2) was provided from a randomized control trial study that examined the acute effects of muscle fatigue on scapular kinematics followed prolonged tennis [30]. The findings of this study suggest that fatigue may lead to subacromial impingement, however, the scapular alterations were examined only within a 24-hour period, suggesting that further research should identify which are the scapular kinematics in a period over 24-hours post-game. Exposure to intensive tennis has been identified as a risk factor for developing degenerative articular changes in the dominant shoulder in tennis players [31]. Despite the absence of statistical analysis resulting in low reliability values, this was the only study that examined shoulder articular changes. The influence of prolonged tennis on shoulder internal range of motion, three hours post-game, was examined in a laboratory environment by Martin and colleagues [20]. Despite the poor sample size (*n* = 8), the findings of this study were similar to the study of Moore-Reed and colleagues [28], confirming that prolonged tennis play is a risk factor for upper limb injury in tennis players.

Further to the above findings, a different study from Martin and the colleagues [29] examined kinematic, kinetic, and performance changes occurring after three hours of play on tennis serve, reporting decreased serve ball speed, decreased maximal knee bend, lower upper limb angular velocities, and decreased upper limb joint kinetics, indicating upper limb muscular fatigue, and indicating that prolonged tennis affects serve biomechanics and may induce fatigue in the upper limb muscle, due to inefficient energy flow, resulting in increased stress of the upper limb and exceeding tissue tolerance, hence causing injury. 

### 4.2. Dyskinesia 

Visible alterations (winging or asymmetry) in the position and the motion of the scapula have been termed scapular dyskinesis [38], responsible for changes in activation of scapular stabilizing muscles [39]. Burn and colleagues [40] reported that overhead athletes have greater prevalence of developing scapular dyskinesis than non-overhead athletes, found to be present in 61% of overhead athletes. Cools and colleagues [22] highlighted the scapular dyskinesis as a possible variable for shoulder pathology in tennis players. Tennis forehand drive might contribute to scapular dyskinesis, according to a study from Rogowski and colleagues [35], mainly due to scapulothoracic anterior tilt width and internal rotation observed during the follow-through phase of the forehand motion. Tennis players with scapular dyskinesis were found to have reduced subacromial space, according to a study from Silva and colleagues [37]. This study used ultrasonographic evaluation to measure subacromial space, confirming the hypothesis that tennis players may develop subacromial impingement. The impact of shoulder and scapular kinematics to shoulder impingement was studied by Lädermann and colleagues [33]. Their findings indicated that nine out of 10 ex-professional tennis players were found to have posterosuperior shoulder impingement, especially when serving, strengthening the theory that this stroke is the most harmful stroke in tennis. The presence of scapular dyskinesis among tennis players was clearly defined in this review and has been identified as a risk factor for upper limb injury. 

### 4.3. Shoulder Rotational Properties

Limited internal rotation (IR), rather than glenohumeral internal rotation deficit GIRD, was associated with shoulder pain history, duration of tennis practice, and player’s age, after comparing shoulder rotation range between professional tennis players with and without shoulder pain history [41]. Shoulder flexibility and shoulder ROM were found to be limited according to a two-year period of study from Kibler and Chandler [23]. This study identified decreased or asymmetrical motion in shoulder internal rotation, without answering how inflexibility could reproduce an injury. Shoulder inflexibility theoretically affects shoulder kinetics, increasing the loading of the joint, and causing a trauma. However, this mechanism is multifactorial, since several key factors such as muscle strength, kinetic chain integrity, shoulder kinematics, quality of strokes, and exposure play a significant role in the pathophysiology of shoulder injury. 

### 4.4. Technique and Skill Level 

Lateral epicondylitis, or “tennis elbow”, was indicated as one of the most common overuse injuries in tennis, especially to novice tennis players [42]. Henning and colleagues [43] suggested that the development of lateral epicondylitis could be associated with poor tennis technique. Nowadays, the incident rate of lateral epicondylitis has been decreased mainly due to improved technical outcomes and by using the two-handed backhand grip [42]. Laboratory studies reported that professional tennis players with a high quality of technique, demonstrated lower shoulder joint kinetics, and therefore theoretically lower shoulder loading, decreasing the risk of overuse injury, showed ability to maximize serve velocities with minimum shoulder kinetic loading [21,29]. An explanation of this decreased risk might be that high-skilled tennis players—despite the increased physiological and biomechanical demands at high competition level—would have a superior ability to transfer forces through the kinetic chain and work more efficiently.

### 4.5. Racquet Properties

Previous research in tennis grip size, string tension, the size of the racquet, and the use of vibration has been proposed to have an effect on the development of lateral epicondylitis [42], while stiffer tennis racquets and lower grip forces reduce the mechanical loads on the arm, without impeding ball velocity [24]. A recent study on the effects of racquet mass modification on upper limb joint loads identified an increased racquet polar moment of inertia. This increase in polar moment of inertia may increase injury risk for tennis players [26], however this study meets several limitations, including poor sample size (*n* = 8) and limited type of serve stroke (the study does not provide details on the type of the serve. According to the study of Hatch and colleagues [32], alterations in tennis racket grip size within ¼ in of Nirschl’s recommended sizing does not significantly affect forearm muscle activity, and therefore may not be considered as a risk factor for lateral epicondylitis, however, playing with a different grip size may alter the quality of the stroke or the kinematics of the wrist joint, resulting in poor technique and therefore increasing the risk of injury. 

Previous biomechanical studies have shown that the non-dominant wrist is in extensive ulnar deviation during the two-handed backhand stroke, increasing the risk for tendon injuries [44]. Ever since, no further study has ever examined extensively any possible connection between the tennis grip patterns and wrist injury mechanisms during the backhand stroke, identifying a serious gap in the literature of tennis medicine. In modern tennis, the forehand stroke is very powerful and mainly used while players are in an open and semi-open stance hitting position and the wrist has a key role in developing angular momentum to increase racquet head speed, especially while players using western or semi western racquet grip [25]. Nowadays, a lot of young tennis players during the forehand stroke are using full western grips with extreme ulnar deviation and elbow valgus increasing the risk for wrist tendon injuries and medial elbow ligament pathology [25]. Different types of grips are related to the anatomic site of the injury according to a study placed in Italy, from Tagliafico and colleagues [25], however the method used in this study to collect the data was based on a questionnaire that was administrated to the players, confirming low quality results since single data report itself does not provide reliable outcomes.

### 4.6. Previous Injury 

Previous injury has been clearly documented as one of the commonest intrinsic factors for musculoskeletal injury in sports [45]. However, this systematic review revealed only one study that examined this hypothesis in tennis, reporting that shoulder injured players demonstrated altered shoulder kinematics and significantly higher shoulder joint kinetics compared to non-injured players [29]. Despite the low sample size (*n* = 20), this was the only study that used different skill level of participants, confirming that previous injury could be considered as a potential risk factor. 

### 4.7. Age

Age-related alterations in glenohumeral and scapulothoracic joints in tennis players have been extensively studied in the past [23,46,47]. Surprisingly, despite the fact that the previous research has clearly identified that shoulder IR deficit increases with age and years of experience, increasing the risk of injury among tennis players, this review failed to confirm this hypothesis, since the only study, which was included in this review, provided limited and conflicted evidence. Cools and colleagues [27] examined how age affects scapular strength and shoulder kinematics using a sample of junior tennis players aged younger than 14 years (*n* = 24), 14 to 16 years (*n* = 22), and older than 16 years (*n* = 13). They applied field measurement tools without any statistical analysis between the variables measured, since the sample in the age category >16 was small, resulting in increasing the risk of bias. Unfortunately, this study could not significantly connect age and injury risk. Future research should investigate the relationship between age and injury risk.

### 4.8. Surface

We would have expected that a different tennis surface may affect quality of play and could be responsible for injuries, however studies in this area are limited only to injuries to lower limbs [48,49] and the trunk [50], and are therefore not included in this review.

## 5. Limitations, Strength, and Practical Applications

The data research could not identify a large number of studies that meet the inclusion criteria, and this is not due to the restrictions applied, but to poor quality of studies, or no studies at all. Surprisingly, previous injury, age, or wrist pathology were not extensively investigated. The quality of studies and level of evidence was low to moderate, despite the fact that the area of the research was not limited to one anatomical area. Furthermore, limitations according to PRISMA framework (e.g., risk of bias and incomplete retrieval of identified research and reporting bias) were acknowledged. On the other hand, strength of the present study was its novelty as it was the first systematic review to identify risk factors for upper limb injury in tennis players. Considering the novelty of this study, the findings would be of great practical relevance for healthcare providers, policy makers, and coaches for the development of injury prevention programs in competitive and recreational tennis players.

## 6. Conclusions

Overall, the quality of the studies included was moderate to low. Prolonged tennis, scapular dyskinesis, muscle fatigue, scapulothoracic properties, shoulder kinetics or kinematics, skill level, and technique were indicated as risk factors for upper limb injury in tennis players. Despite the fact that age and previous injury have been extensively studied in the past, there is a need for higher quality research to examine their connection to tennis injuries. We could not identify studies examining overloading or workloads as a risk factor for tennis injuries that met the inclusion criteria. We assume that overloading and training loads have not well-studied so far in tennis due to difficulty to apply such measures. It is likely that the advancement of technology (GPS, accelerometry) will help in future studies following existing research in other sports (e.g., soccer, rugby). Tennis needs validated workload monitoring methods to examine the behavior of loads during training and matches and for identifying a potential connection to injury. The necessity of more epidemiological studies from different age and performance groups of players and different skill levels is highly recommended and further research is needed to examine the mechanism behind tennis injuries. Finally, identifying risk factors for upper limb injury in tennis players would benefit clinicians, sport scientists, and coaches to design an effective injury prevention strategy and enhance performance.

## Figures and Tables

**Figure 1 ijerph-17-02744-f001:**
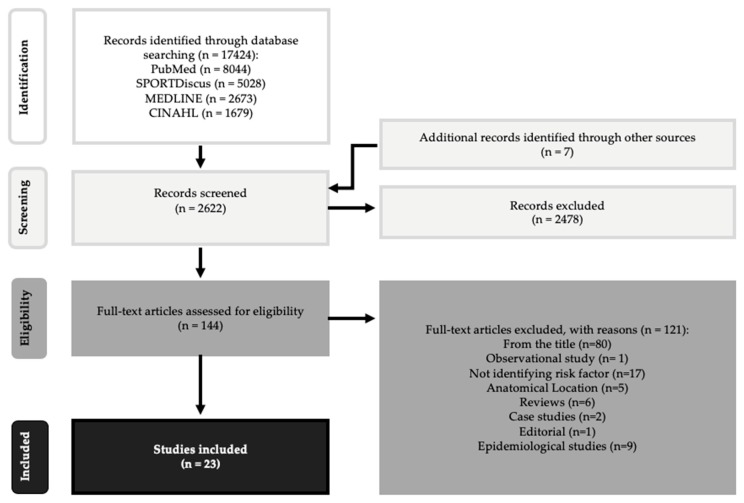
Flow chart of the selection process according to PRISMA statement [15].

**Table 1 ijerph-17-02744-t001:** Overview of the database search.

		Pubmed	Sportdiskus	Medline	Cinahl
Age AND tennis	Exclusion criteria search	758	1498	752	447
	Inclusion criteria search	477	448	98	74
Tennis Injuries	Exclusion criteria search	2242	598	118	92
	Inclusion criteria search	1087	143	6	9
Muscle AND tennis	Exclusion criteria search	839	889	683	330
	Inclusion criteria search	504	233	64	49
Elbow injuries AND tennis	Exclusion criteria search	1613	141	70	113
	Inclusion criteria search	787	28	1	9
Shoulder Kinetics OR kinematics AND tennis	Exclusion criteria search	348	329	145	132
	Inclusion criteria search	237	65	23	30
Kinematics AND tennis	Exclusion criteria search	344	257	95	78
	Inclusion criteria search	233	90	38	34
Kinetics AND tennis	Exclusion criteria search	157	369	156	41
	Inclusion criteria search	124	63	8	13
Injury risk AND Tennis	Exclusion criteria search	283	103	77	49
	Inclusion criteria search	190	21	7	9
Shoulder Injury AND tennis	Exclusion criteria search	258	134	41	67
	Inclusion criteria search	153	17	2	5
Overuse Injury AND tennis	Exclusion criteria search	233	139	61	42
	Inclusion criteria search	114	22	2	7
Risk factors AND tennis	Exclusion criteria search	221	69	117	67
	Inclusion criteria search	151	11	8	9
Shoulder Rotation AND tennis	Exclusion criteria search	118	102	65	42
	Inclusion criteria search	83	14	8	8
Wrist Injuries AND tennis	Exclusion criteria search	91	55	39	27
	Inclusion criteria search	41	7	0	3
Glenohumeral AND tennis	Exclusion criteria search	45	68	45	31
	Inclusion criteria search	30	8	1	2
Asymmetry AND tennis	Exclusion criteria search	60	60	43	15
	Inclusion criteria search	45	19	7	7
Scapular AND tennis	Exclusion criteria search	49	36	49	28
	Inclusion criteria search	33	6	5	7
Scapula AND tennis	Exclusion criteria search	48	28	44	31
	Inclusion criteria search	30	5	3	5
Shoulder Kinematics AND tennis	Exclusion criteria search	107	12	6	5
	Inclusion criteria search	76	4	0	2
Overhead Injuries AND tennis	Exclusion criteria search	68	20	13	16
	Inclusion criteria search	49	7	3	5
Training loads AND tennis	Exclusion criteria search	26	74	8	4
	Inclusion criteria search	23	22	5	0
Kinetic Chain AND tennis	Exclusion criteria search	30	26	20	11
	Inclusion criteria search	27	8	5	3
Dyskinesis AND Tennis	Exclusion criteria search	48	7	11	4
	Inclusion criteria search	40	3	3	1
Dyskinesia AND Tennis	Exclusion criteria search	40	1	8	1
	Inclusion criteria search	34	0	0	0
Shoulder Kinetics AND Tennis	Exclusion criteria search	18	13	7	6
	Inclusion criteria search	15	2	0	1

**Table 2 ijerph-17-02744-t002:** Summary of the characteristics of the included studies.

Study	Type of Study	Participants	Sample Size	Risk Factor Examined
[22]	Cross-sectional study	Competitive junior tennis players	N = 35, (M = 19, F = 16)	Scapulothoracic position, muscle strength, flexibility
[27]	Cross-sectional study	Competitive adult tennis players	N = 59, (M = 31, F = 28)	Age-related shoulder/scapular adaptions
[32]	Controlled laboratory study	Collegiate tennis players	N = 16’ (Μ)	Racket grip size
[24]	Cross-sectional study	Competitive adult tennis players	N = 55, (Μ)	Racket properties
[13]	Prospective 2-year study	Competitive junior tennis players	N = 55, (M = 35, F = 20)	Previous injury
[23]	Laboratory-based study	Competitive junior tennis players	N = 51, (M = 29, F = 22)	Flexibility and range of motion
[33]	Laboratory study	Ex-professional senior tennis players	N = 10, (M = 9, F = 1)	Glenohumeral instability and shoulder impingement
[21]	Cross-sectional study	Professional and competitive tennis players	N = 18, (M = 18)	Skills and technique
[34]	Laboratory-based study	Competitive adult tennis players	N = 20, (M = 20)	Skill, technique kinetic chain, and previous injury
[29]	Laboratory-based study	Competitive adult tennis players	N= 8, (M = 8)	Effect of prolonged tennis to shoulder muscle fatigue
[20]	Laboratory-based study	Competitive adult tennis players	N= 8, (M = 8)	Effect of prolonged tennis to shoulder range of motion
[31]	Cross-sectional controlled study	Competitive senior tennis players	N = 18, (M = 17, F = 1)	Prolonged tennis may affect shoulder articular cartilage
[28]	Cohort study	Professional tennis players	N = 79, (F = 79)	Effect of prolonged tennis on glenohumeral rotation
[30]	Randomized controlled clinical trial	Collegiate tennis players	N = 20, (M = 20)	Influence of fatigue on scapular kinematics
[35]	Laboratory-based study	Competitive adult tennis players	N = 8, (M = 8)	Scapulothoracic kinematics
[26]	Laboratory-based study	Competitive adult tennis players	N = 8, (M = 8)	Racket polar moment of inertia
[36]	Cross-sectional study	Competitive junior tennis players	N = 40, (M = 26, F = 14)	Shoulder rotational muscle imbalances
[37]	Cross-sectional study	Competitive junior tennis players	N = 53, (M = 31, F = 22)	Correlation between scapular dyskinesia and subacromial space
[25]	Cross-sectional study	Competitive adult tennis players	N = 400, (M = 323, F = 77)	Racket grip

N = number of participants; M = male participants; F = female participants.

**Table 3 ijerph-17-02744-t003:** Summary of the checklist for measuring quality of study reporting.

Studies	Clear Description of the Hypothesis, Objectives	Outcomes Clearly Described in the Introduction or Methods Section	Clear Description of the Patient’s Characteristics	Clear Description of Intervention of Interest	Clear Description of the Distribution of Principal Confounders	Clear Description of Study Findings	Estimates of the Random Variability in the Data for the Outcomes	Measurement of Adverse Events	Description of Patient’s Characteristics that were Lost to Follow-Up	Report of Probability Values	Score
[22]	1	1	1	1	0	1	1	0	0	1	7/10
[27]	1	1	1	1	0	1	1	0	0	1	7/10
[32]	1	1	1	1	0	1	1	0	0	0	6/10
[24]	1	0	0	0	0	0	0	0	0	0	1/10
[13]	1	0	1	1	0	1	1	0	0	1	6/10
[23]	0	0	1	1	0	0	1	0	0	1	4/10
[33]	1	1	0	1	0	1	1	0	0	1	6/10
[21]	1	1	1	1	0	1	1	0	0	1	7/10
[34]	1	1	1	1	0	1	1	0	0	1	7/10
[29]	1	1	1	1	0	1	1	0	0	1	7/10
[20]	1	1	0	1	0	1	1	0	0	1	6/10
[31]	1	1	1	1	0	1	1	0	0	1	7/10
[28]	1	1	1	1	0	1	1	0	0	1	7/10
[30]	1	1	1	1	0	1	1	0	0	1	7/10
[35]	1	1	1	1	0	1	1	0	0	1	7/10
[26]	1	1	1	1	0	1	1	0	0	1	7/10
[36]	1	1	1	1	0	1	1	0	0	1	7/10
[37]	1	1	1	1	0	1	1	0	0	1	7/10
[25]	1	1	1	1	0	1	1	0	0	1	7/10

**Table 4 ijerph-17-02744-t004:** Summary of the checklist for measuring external validity.

Studies	Were the Subjects Asked to Participate in the Study Representative of the Entire Population from which they were Recruited?	Were Those Subjects who were Prepared to Participate Representative of the Entire Population from which they were Recruited?	Were the Stuff, Places, and Facilities where the Patients were Treated Representative of the Treatment the Majority of Patients Received?	Score
[22]	0	1	1	2/3
[27]	0	1	1	2/3
[32]	0	1	1	2/3
[24]	0	1	1	2/3
[13]	0	1	1	2/3
[23]	0	1	1	2/3
[33]	1	1	1	3/3
[21]	1	1	1	3/3
[34]	1	1	1	3/3
[29]	1	1	1	3/3
[20]	0	1	1	2/3
[31]	0	1	1	2/3
[28]	0	1	1	2/3
[30]	0	1	1	2/3
[35]	1	1	1	3/3
[26]	1	1	1	3/3
[36]	0	1	1	2/3
[37]	1	1	1	3/3
[25]	0	1	1	2/3

**Table 5 ijerph-17-02744-t005:** Summary of the checklist for measuring internal validity (risk of bias).

Studies	Was an Attempt Made to Blind Study Subjects to the Intervention they had Received	Was an Attempt Made to Blind those Measuring the Main Outcomes of the Intervention	If any of the Results of the Study were Based on “Data Dredging”, was this Made Clear?	In Trials/Cohort Studies, do the Analyses Adjust for Different Lengths of Follow-Up of Patients, or in Case Control Studies, is the Time Period between the Intervention and Outcome the Same for Cases and Controls	Were the Statistical Tests Used to Assess the Main Outcomes Appropriate	Was Compliance with the Intervention/s Reliable?	Were the Main Outcome Measures Used Accurate (Valid and Reliable)?
[22]	0	0	0	1	1	0	1
[27]	0	0	0	1	0	0	1
[32]	0	0	0	1	0	0	1
[24]	0	0	0	1	0	0	1
[13]	0	0	0	1	1	0	1
[23]	0	0	0	1	0	0	1
[33]	0	0	0	1	1	0	1
[21]	0	0	0	1	1	0	1
[34]	0	0	0	1	1	0	1
[29]	0	0	0	1	1	0	1
[20]	0	0	0	1	0	0	1
[31]	0	0	0	1	1	0	1
[28]	0	0	0	1	1	0	1
[30]	0	0	0	1	1	0	1
[35]	0	0	0	1	1	0	1
[26]	0	0	0	1	1	0	1
[36]	0	0	0	1	1	0	1
[37]	0	0	0	1	1	0	1
[25]	0	0	0	1	1	0	1

**Table 6 ijerph-17-02744-t006:** Summary of the checklist for measuring internal validity-confounding (selection bias).

Studies	Were the Patients in Different Intervention Groups or were the Cases and Control Recruited from the Same Population	Were Study Subjects in Different Intervention Groups (Trials and Cohort Studies) or were the Cases and Controls Recruited over the Same Period	Were Study Subjects Randomized to Intervention Groups	Was the Randomized Intervention Assignment Concealed from Both Patients and Health Care Staff until Recruitment was Complete and Irrevocable?	Was there Adequate Adjustment for Confounding in the Analysis from which the Main Findings were Drawn?	Were Losses of Patients to Follow-Up Taken into Account
[22]	0	0	0	0	0	0
[27]	0	0	0	0	0	0
[32]	0	0	0	0	0	0
[24]	0	0	0	0	0	0
[13]	0	0	0	0	0	0
[23]	0	0	0	0	0	0
[33]	0	0	0	0	0	0
[21]	0	0	0	0	0	0
[34]	0	0	0	0	0	0
[29]	0	0	0	0	0	0
[20]	0	0	0	0	0	0
[31]	0	0	0	0	0	0
[28]	0	0	0	0	0	0
[30]	1	0	1	1	0	0
[35]	0	0	0	0	0	0
[26]	0	0	0	0	0	0
[36]	0	0	0	0	0	0
[37]	0	0	0	0	0	0
[25]	0	0	0	0	0	0

**Table 7 ijerph-17-02744-t007:** Summary of injury risk factors by quantity, quality, and level of evidence.

Studies	Level of Evidence	Level 1	Level 2			Level 3		
	Risk Factor		RCT Study	Cohort Study	Cross-Sectional Study	Laboratory Study	Prospective Study	Studies
[22]	Scapulothoracic position				12 (26)			1
[22]	Muscle strength				12 (26)			1
[22]	Shoulder flexibility				12 (26)			2
[23]						8 (26)		
[27]	Age-related shoulder/scapular adaptions				11 (26)			1
[32]	Racquet grip size					10 (26)		1
[24]	Racquet properties					5 (26)		1
[13]	Prolonged tennis						11 (26)	5
[29]	Effect of prolonged tennis to shoulder muscle fatigue					13 (26)		
[20]	Effect of prolonged tennis to shoulder range of motion					10 (26)		
[31]	Prolonged tennis may affect shoulder articular cartilage				12 (26)			
[28]	Effect of prolonged tennis on glenohumeral rotation			14 (26)				
[13]	Previous injury						11 (26)	2
[34]						13 (26)		
[23]	Range of motion					8 (26)		1
[33]	Glenohumeral instability					12 (26)		1
[21]	Skills					13 (26)		2
[34]						13 (26)		
[21]	Technique					13 (26)		2
[34]						13 (26)		
[34]	Kinetic chain					13 (26)		1
[30]	Influence of fatigue on scapular kinematics		12 (26)					1
[35]	Scapulothoracic kinematics					13 (26)		1
[26]	Racquet polar moment of inertia					13 (26)		1
[36]	Shoulder rotational muscle imbalances				12 (26)			1
[37]	Correlation between scapular dyskinesia and Subacromial space				13 (26)			1
[25]	Racquet grip				12 (26)			1

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
