# Peer review of "Risk Factors for Upper Limb Injury in Tennis Players: A Systematic Review"

_ijerph, 2020, doi:10.3390/ijerph17082744_

Round 1

Reviewer 1 Report

Introduction

L36 - Perhaps remove (three different surfaces) or clarify for Grand slam events. There are three major surfaces but also a number of additional surfaces, ie indoor surfaces - carpet (synthetic grass), other outdoor versions of clay - et tout cas (and so on).

Author Response

Reply to Reviewer 1

Comment:

Introduction

L36 - Perhaps remove (three different surfaces) or clarify for Grand slam events. There are three major surfaces but also a number of additional surfaces, ie indoor surfaces - carpet (synthetic grass), other outdoor versions of clay - et tout cas (and so on).

Answer: We agree with the expert reviewer and revised this part to “Tennis differs from other sports in terms of match duration (exposure), surface of play and equipment [2] ”

Reviewer 2 Report

The changes were performed accordingly. However:

CONCLUSIONS
MAJOR 06: pg. 16 – lines 11-13. “Surprisingly, there were no studies examining overloading or
training load as a risk factor for tennis injuries that met the inclusion criteria.” For what reason would
this finding has happened?
Answer: We assume that overloading and training loads have not well-studied so far in tennis about
the difficulty to apply such measures. Likely, the advancement of technology (gps, accelerometry) will
help in future studies following existing research in other sports (e.g. soccer).

R2: Add this information in the manuscript.

Author Response

Replying to reviewer 2

Comment:

CONCLUSIONS
MAJOR 06: pg. 16 – lines 11-13. “Surprisingly, there were no studies examining overloading or
training load as a risk factor for tennis injuries that met the inclusion criteria.” For what reason would
this finding has happened?

Answer: We agree with the expert reviewer and revised this conclusion (“We assume that overloading and training loads have not well-studied so far in tennis due to difficulty to apply such measures. Likely, the advancement of technology (GPS, accelerometry) will help in future studies following existing research in other sports (e.g. soccer, rugby).”)

This manuscript is a resubmission of an earlier submission. The following is a list of the peer review reports and author responses from that submission.

Round 1

Reviewer 2 Report

GENERAL COMMENTS REGARDING PAPER

This study has the well done theoretical and practical approaches and it can be instructional for teachers and researchers.

MAJOR COMPULSORY REVISIONS METHODS

ABSTRACT

MAJOR 01: pg. 1 – lines 26-28. Sounds like similar the last sentence.

MAJOR 02: pg. 1 – line 31. To increase the chances of impact of the study, the keywords should be different from the words used in the title.

RESULTS

MAJOR 03: pg. 6 – lines 29-30. Please, insert one section with data on “Characteristics of the Studies”. For example, “pooled sample was xxxx participants (xx% male, xx ± x years old; xx% female, xx ± x years old). xx% of them were professional, xx% youth academy, xx% collegiate...” and other relevant information.

MAJOR 04: pg. 6 – line 21. What’s “prolonged tennis”?. All “Key findings-risk factors “ should be defined and clearer to the reader.

MAJOR 05: pg. 8-13. A lot of tables impairing the flow and many of them can be put as “Additional files”.

 CONCLUSIONS

MAJOR 06: pg. 16 – lines 11-13. “Surprisingly, there were no studies examining overloading or training load as a risk factor for tennis injuries that met the inclusion criteria.” For what reason would this finding has happened?

Reviewer 3 Report

General Overall Comments:

Positives

Analysing a large number of studies which investigate risk factors in sport can assist health providers to develop comprehensive treatment and management plans. In combination with previous research on incident rate, location and type of injury, this review could also form part of a larger study to potentially inform athlete development policies.

Weaknesses

A meta-analysis should include forest plots, illustrating effect size estimates and confidence intervals for all included studies. Downs and Black criteria has been used to measure quality, which is represented in Tables 3- 6; however, power calculation has not been included (or not clear). Could clarify? PRISMA framework also suggests that outcomes should be included for each study, i.e., simple summary data for each intervention group, effect estimates and confidence intervals, ideally with a forest plot.

Referring to PICOS methodology, the population could be further defined, e.g., authors could include studies on elite youth tennis players or senior recreational tennis players only. There can be significant variability within a population of tennis players.

Although identified in the Introduction, court surface was not included in the database search. Court surface can have an impact on technique and load on the wrist and shoulder joint. Therefore, it is possible that some relevant studies have been missed.

Overall, the sample sizes for the included studies were relatively small.

Geographical differences in the included studies (variables in coaching/playing styles, surface, access to resources, socioeconomic factors) could have an impact on injury risk factors and should be acknowledged.

 Ideally, the results of the twenty different studies should be homogenous. As the tennis population can have large variations, heterogeneity may exist and therefore authors should report and estimate if significant (include Chi-squared test).

Terminology also needs to be clarified, i.e., match duration was defined as tennis exposure, but how does prolonged tennis and intensive tennis differ? Participant characteristics (Table 2) also needs clarification, i.e., senior tennis players, ex-professional (could be also senior), college, elite junior, etc.

Specific Comments

Abstract

L 23 - PRISMA framework suggests a statement of questions should be addressed with reference to participants, interventions, comparisons, outcomes, and study design (PICOS).

 Clarify PICO or PICOS methodology and clarify participants, ie., is tennis players too broad?

Systematic review registration number should be included

 Introduction

L 34 – Opening sentence could be clarified, suggests anyone that plays tennis is at risk of injury?

Pg. 2, L 3 – Clarify definition of senior tennis players, i.e., ex professional, recreational.

L 13 – Tendinopathy also depends on other variables, i.e., skill and technique, age, load. Not clear if this is referring to study at US Open, if so, relates to professional tennis players, Hard Court surface, extreme playing conditions, prolonged tennis (closer to end of season).

L 22 – Is there information missing? Doesn’t quite flow to the next section, leaves the kinetic chain information hanging.

Methods (Experimental Section)

L 34 – Was PubMed included in the database search? PubMed has been included in Fig 1 and Table 1. EMBASE has been listed but not included in Fig 1 and Table 1. Spelling of SportDiscus – also in Table 1.

Pg. 5 L2 – Perhaps database search could be expanded to include tennis court surface AND injury, tennis equipment AND injury. Potential for some relevant studies to have been missed.

L2 - Table 2 suggests three studies relating to racket properties have been included in the systematic review (but not explicitly stated in database search).

L 9 - Technology has certainly changed the game of tennis, good that only studies since 2000 have been included (could perhaps even refine this to the last 10 years).

L 21 – Mentions power has been reported but not included in Tables (or not clear). According to Downs and Black quality assessment tool power should be reported.

L 21 – Should quality and level of evidence for each study be represented in forest plot?

Results

Pg. 6 L 4 – Suggests characteristics of the studies included are summarised in Table 2. Referring to PRISMA framework the characteristics for each study should also include PICOS methodology, follow-up period and citations. As noted above, results of individual studies should include (a) simple summary data for each intervention group (b) effect estimates and confidence intervals, ideally with a forest plot.

L 6 – 7 – Participation sample involved tennis players from different ages, gender and skill levels. This suggest a level of heterogeneity may exist; therefore, authors should report the significance (chi-squared test). However, in L 8 it is suggested that only one skill level was included, this needs to be clarified in Table 2. For example, specify the difference in skill level between (a) adult tennis players (b) senior tennis players (c) competition adult players (d) non-professional competitive tennis players (e) competitive tennis players (f) expert, intermediate and recreational tennis players etc.

L 15 - Checklists for measuring quality of studies is listed in Tables 3 – 5. However, as noted above, power should be reported as an element of Downs and Black quality assessment tool, missing from Table 5?

L 21 – Further clarify prolonged tennis and how this differs from exposure to tennis?

Discussion

L 44 – Skills and technique have been included together throughout review, should also be here.

L 50 – Reported that three studies used professional tennis players. Perhaps discuss how this may have an impact on overall review results, i.e., professional players will report an increase in physical demands and biomechanical loading, but reduced force on upper body due the understanding and ability to transfer forces through the kinetic chain. Details have been mentioned in pg. 14 L 42 – but could be further highlighted.

Pg. 7, L 2 - Summary of evidence. Consider the relevance to key groups (e.g., healthcare providers, policy makers and coaches).  As noted above, clarity of terms - prolonged, exposure and intensive tennis may strengthen evidence and provide greater appeal to a larger audience.

L 28 – Should previous injury study be included in the review? Only one study with small sample size.

L 46 – Discuss limitations according to PRISMA framework. Further discuss limitations at study and outcome level (e.g., risk of bias), and at review-level (e.g., incomplete retrieval of identified research, reporting bias). Not initially clear that limitations are referring to overall systematic review. 

Conclusion

L 16 – Has provided a good general interpretation of the results in the context of other evidence, and implications for future research. Relevant to clinicians and sports scientist and could also be relevant to coaches and policy makers if terminology is clarified.
